# Structural layer applicability of semi-flexible material for rutting resistance: A coupled temperature-mechanical approach

**Maohua Yu[1], Tianming He[1], Kejian Xu[2], Hong Cheng[3], Minda Ren[4]***

**1** Xinzhou Highway Development Center, Shangrao, Jiangxi, China, **2** Yugan Highway Development Center, Shangrao, Jiangxi, China, **3** Poyang Highway Development Center, Shangrao, Jiangxi, China, **4** Key Laboratory of Civil Engineering Structure and Mechanics, Inner Mongolia University of Technology, Hohhot, Inner Mongolia, China

* m_ren@imut.edu.cn

**Data Availability Statement:** All relevant data are within the paper and its Supporting Information files.

## Abstract

Semi-flexible material (SFM) is produced by pouring cement grouting material into the asphalt concrete skeleton. It exhibits both characteristics of cement and asphalt, increasing structural stiffness and reducing rutting. Extensive studies have shown that the temperature load coupling effect is one of the leading causes of road rutting. However, few researchers focused on the anti-rutting impact and structural layer applicability of SFM under this effect. Thus, a coupled temperature-mechanical approach was developed based on the finite element (FE) method to simulate the rutting of SFM at different pavement layers and times of the day. During simulation, both standard load and overload were applied to the FE model of pavement. Asphalt mixture and SFM specimens were prepared for essential road performance and dynamic modulus testing. The mechanical properties of SFM and asphalt mixtures at different temperatures were obtained based on the measured data. The structural layer applicability of SFM was revealed by simulating the response of the pavement structure under the combined action of temperature and load. An accelerated pavement test (APT) based validation indicated that the simulation results were accurate. The results show that traditional asphalt pavement and pavement with SFM at the surface and bottom layers tend to exhibit dilative heave adjacent to the wheel load. Using SFM at the middle layer shows a compacted rutting mode, and the pavement has a minimum rise of 51% in rutting depth under the double overloading compared with the pavements with SFM in other layers. It implies that using SFM in the middle layer gives optimal resistance to overload. Considering the depth, form, and resistance of rutting, the SFM in the middle layer of pavement can functionally exert its anti-rutting characteristic.

## 1 Introduction

Ruts have been one of the most common diseases on asphalt pavement, seriously affecting structural durability and driving safety [1, 2]. In fields, the surface layer is subject to the

**Funding:** The authors acknowledge the support received from the Jiangxi Provincial Department of Transportation Project (NO. 2022SF003).

**Competing interests:** We declare that no support, financial or otherwise, has been received from any organization that may have an interest in the submitted work and there are no other relationships or activities that could appear to have influenced the submitted work.

combined action of environment and load, and asphalt mixture, as a typical rheological material, its properties change with temperature. Thus, when studying road rutting, it is essential to focus on the coupling effect of temperature load [3, 4].

Currently, the prevention of rutting mainly consists of two aspects: optimization of structural combinations and development of anti-rutting materials [5]. Researchers have achieved the effects of reinforcement and shear resistance by applying geo-grids to road surfaces [6]. In addition, many studies have focused on improving the high-temperature deformation resistance of asphalt mixtures, including adding polymers, fibers, rubber, etc., to improve the overall stiffness of asphalt mixtures and reduce rutting deformation [7]. However, the application results indicate that the improvement effect is not stable, mainly due to the strong temperature dependence of the asphalt mixture itself. Although some methods have improved the high-temperature deformation resistance, it challenges researchers to eliminate the asphalt mixtures' flexible nature.

The semi-rigid and semi-flexible pavement material is formed by injecting a particular cement slurry into a large void asphalt mixture, which has the superior properties of both asphalt and cement concrete. Studies have shown that SFM can effectively increase the overall stiffness of pavement structures and alleviate the diseases caused by the surge of traffic [8–11]. It is recognized that rutting or pushing diseases on asphalt roads are caused by traffic conditions (such as traffic density, heavy load, and slow traffic) and extremely high temperatures of pavement structure [5, 12]. Studies have found that although the overall dynamic modulus of SFM is less sensitive to temperature, there are significant differences in their internal component's temperature responses. In addition, with the interlocking effect of aggregate structure, the internal structure formed by cement slurry plays a vital role in providing strength for SFM composite materials, especially under high-temperature conditions [13]. Based on this, each component's contribution to overall strength will vary with temperature. Thus, the coupling effect of temperature load is the critical factor in revealing the rutting resistance of SFM. However, extensive studies focused on the resistance of SFM to heavy vehicles and static loads. Many traditional test methods, such as indirect tensile testing, low-temperature bending test, freeze-thaw test, and saturated marshall test, are used to evaluate the performance of SFM under different failure conditions [9]. Using such tests challenges researchers to determine the specific impact of SFM on structural response and reveal the mechanical evolution law under the action of load temperature coupling.

Numerical simulation methods can calculate the mechanical response of SFM under the influence of internal field variables (that are not easy to be obtained through traditional experiments) by defining field variables and their correlation functions. Cai et al. [13] quantified the role of temperature in the mesoscale damage process of semi-flexible pavement composite materials using the FE method. They found that temperature changes significantly impact the damage evolution of various phases and interfaces inside SFM. The inherent characteristics of SFM will lead to different strain behaviors of their internal components under the same stress [14]. The FE method can be used to visualize and quantify the stress and strain distribution inside the material. Ding et al. [15] studied the mechanical behavior and failure mechanism of semi-flexible pavement materials containing RAP at different scales. They found that the failure probability of the asphalt phase inside SFM was the highest at room temperature, and the failure probability of the original aggregate was the lowest. Jiang et al. [16] built a mesostructure model of SFM, analyzed the principal stress and shear stress of different phases in the material, and found that the internal stress concentration occurred at the sharp points of the base asphalt mixtures. Many studies have used the FE method to analyze the internal mechanical response of SFM at the microscopic scale. Few studies have considered the temperature-

**Table 1. Technical indexes of matrix asphalt.**

| Technical index | | Unit | Test data | Technical requirement |
|---|---|---|---|---|
| Penetration (25˚C, 100g, 5s) | | 0.1mm | 71 | 60~80 |
| Penetration index | | | -1.3 | -1.5~+1.0 |
| Ductility (10˚C, 5cm/min) | | cm | 26 | ≥20 |
| Softening point | | ˚C | 52 | ≥46 |
| Flash point | | ˚C | 291 | ≥260 |
| Solubility | | % | 99.7 | ≥99.5 |
| Density (15˚C) | | g/cm$^3$ | 1.048 | Actual measurement |
| TFOT or RTFOT | Mass loss | % | -0.1 | ±0.8 |
| | Residual penetration ratio | % | 72 | ≥61 |
| | Residual ductility (10˚C) | cm | 10 | ≥6 |

load coupling effect to investigate the mechanical response of SFM in the macro pavement structure.

Extensive studies have revealed the deformation of asphalt mixtures under temperature load coupling and have proposed many effective measures to alleviate the occurrence of rutting. However, current prevention methods are challenging to eliminate the flexible nature of traditional asphalt-based materials. The emergence of SFM provides an effective alternative, but its applicability in structural layers is unspecified. To address this issue, we used the FE method to build a road structure model, and the road surface deformation with SFM at different pavement structure layers was calculated. During the simulation, a coupled temperature and load approach was used. The influence of temperature and overload on the anti-rutting effect of SFM was analyzed, and the optimal asphalt layer of anti-rutting of SFM was revealed.

## 2 Materials, specimens, and tests

The asphalt mixture is currently the most widely used material in pavement construction. Its rheological properties dictate the temperature-dependent essence. To improve the mechanical performance of the asphalt mixture, researchers developed SFM, inheriting both advantages of asphalt and cement. This section introduced the tests and measured properties of materials.

### 2.1 Raw material properties

The basic properties of matrix asphalt and SBS-modified asphalt are shown in Tables 1 and 2. The technical requirements refer to the Technical Specification for Construction of Highway

**Table 2. Technical indexes of SBS-modified asphalt.**

| Technical index | | Unit | Test data | Technical requirement |
|---|---|---|---|---|
| Penetration (25˚C, 100g, 5s) | | 0.1mm | 57 | 40~60 |
| Ductility (5˚C, 5cm/min) | | cm | 29 | ≥20 |
| Dynamic viscosity (135˚C) | | Pa·s | 2.4 | ≤3 |
| Softening point | | ˚C | 77 | ≥60 |
| Flash point | | ˚C | 292 | ≥230 |
| Solubility | | % | 99.8 | ≥99 |
| Density (15˚C) | | g/cm$^3$ | 1.023 | Actual measurement |
| TFOT或RTFOT | Mass loss | % | -0.3 | ±1.0 |
| | Residual penetration ratio | % | 72 | ≥65 |
| | Residual ductility (5˚C) | cm | 17 | ≥15 |

**Table 3. Basic properties of cement-based grouting materials.**

| mobility (s) | Bending strength (MPa) | | Compressive strength (MPa) | | Setting time (min) | | 28 days drying shrinkage (%) |
|---|---|---|---|---|---|---|---|
| | 7 days | 28 days | 7 days | 28 days | Initial setting | Final setting | |
| 10.5 | 4.8 | 5.8 | 29.4 | 40.1 | 179 | 328 | 0.14 |

Asphalt Pavements (JTG F40-2004). The cement-based grouting material used in this article is a mixture of cement and water, with cement produced by Taicang Conch Cement Co., Ltd. When the ratio of cement to water is 0.5, the basic properties of cement-based grouting material are shown in Table 3. Basalt is selected for coarse aggregate, and limestone for fine aggregate. The basic properties are shown in Tables 4 and 5.

## 2.2 Design of dense graded asphalt mixture

The surface layer asphalt mixture commonly uses SBS-modified asphalt to ensure high-temperature stability and durability. The middle and bottom layer asphalt mixture can use neat asphalt to reduce costs. Therefore, in this paper, SBS-modified asphalt was used to prepare the SUP-13 mixture specimen and SFM-20 specimen, and matrix asphalt was used to prepare SUP-20 and SUP-25 mixture. The asphalt skeleton of SFM adopts Open Graded Friction Course (OGFC-20), and Table 6 summarizes the design gradation of these four mixtures.

According to the gradation shown in Table 6, the optimal asphalt dosage test was conducted, and the optimal asphalt dosage for SUP-13 was determined to be 4.8%, SUP-20 4.3%, and SUP-25 4.0%. OGFC-20 was designed with 25% void, and the optimal asphalt content was 3.8% by calculation. We prepared the SFM specimens by pouring cement slurry into the OGFC-20 specimens manufactured by the rotary compaction and rolling wheel methods. After pouring and vibrating, the specimens were placed indoors at a temperature of 25°C and a relative humidity of 90% for 14 days for subsequent testing.

## 2.3 Basic performance test of asphalt mixtures

To evaluate the field performance of SUP-13, SUP-16, SUP-20, and SFM mixtures, rutting tests, low-temperature small beam bending tests, and freeze-thaw cycle tests were conducted on the above mixtures. All relevant tests were conducted per the requirements of Standard Test Methods of Bitumen and Bituminous Mixtures for Highway Engineering (JTG E20-2011) [17]. Each test should have at least three specimens, and the specific size and conditions of the test specimens are shown in Table 7.

## 2.4 Dynamic compression modulus test

According to the requirements of the Design Specification for Highway Asphalt Pavements (JTG D50-2017), when calculating the deflection value for road surface acceptance, the

**Table 4. Technical indexes of coarse aggregate quality.**

| Technical index | Unit | Test data | Technical requirement |
|---|---|---|---|
| Crush value | % | 15.4 | $\leq 24$ |
| Los Angeles wear loss | % | 13.9 | $\leq 26$ |
| Apparent density | g/cm$^3$ | 2.858 | $\geq 2.6$ |
| Adhesion to asphalt | level | 5 | 5 |
| Needle flake content | % | 8.9 | $\leq 12$ |
| Solidity | % | 8 | $\leq 12$ |

**Table 5. Technical indexes of fine aggregate quality.**

| Technical index | Unit | Test data | Technical requirement |
|---|---|---|---|
| Apparent relative density | g/cm$^3$ | 2.738 | $\geq$2.5 |
| Silt content | % | 2.2 | $\leq$3 |
| Sand equivalent | % | 74 | $\geq$60 |
| Methylene blue value | g/kg | 5 | $\leq$25 |
| Angularity | s | 40 | $\geq$30 |

dynamic compression modulus of the asphalt surface layer is used at 20°C and 10Hz. However, since the temperature in the actual road structure decreases with the increase of the thickness of the structural layer, the modulus values corresponding to different temperatures should be used in the temperature load coupling analysis to calculate the mechanical response of the pavement under the field environment. Therefore, the test temperatures of 20°C, 30°C, 40°C, 50°C and 60°C are selected in this section. According to the requirements of AASHTO TP79-15 [18], a haversine waveform is used for loading, and the strain range applied to the specimen is 50$\mu\varepsilon$-115$\mu\varepsilon$. When the cumulative plastic deformation of the specimen at various frequencies exceeds 1500, the test is terminated. During the test, the specimen is subjected to a temperature range from low to high (20°C→60°C), with a two-minute interval. The load and deformation data of the five waveforms before the test termination are recorded for calculating the dynamic modulus, as shown in Fig 1.

According to the stress-strain relationship under loading, the complex modulus of asphalt mortar can be obtained, as shown in Eq (1):

$$E^* = \frac{\sigma(t)}{\varepsilon(t)} = \frac{\sigma_0 e^{i(\omega t + \varphi)}}{\varepsilon_0 e^{i\omega t}} \tag{1}$$

where $E^*$ represents the complex modulus, MPa; $\sigma(t)$, $\varepsilon(t)$ respectively represent the stress and strain responses during loading; $\sigma_0$, $\varepsilon_0$ respectively represent the peak values of stress and strain responses, with stress units in MPa and non-dimensional strain; $\omega$ represents angular frequency, rad/s; $\varphi$ represents the phase angle, which can be calculated according to Eq (2).

$$\varphi = \frac{t_i}{t_p} \times 360 \tag{2}$$

where $t_i$ represents the average lag time of the five waveforms before the termination of the test; $t_p$ represents the period of 5 waveforms before the termination of the experiment.

The dynamic modulus is the absolute value of the complex modulus, which is usually calculated using the ratio of peak stress to peak strain in the test, as shown in Eq (3):

$$|E^*| = \frac{\sigma_0}{\varepsilon_0} \tag{3}$$

**Table 6. Composition of four mixtures.**

| Gradation types | Percentage passing through sieve aperture (%) | | | | | | | | | | | | |
|---|---|---|---|---|---|---|---|---|---|---|---|---|---|
| | 31.5 | 26.5 | 19 | 16 | 13.2 | 9.5 | 4.75 | 2.36 | 1.18 | 0.60 | 0.30 | 0.15 | 0.075 |
| SUP-13 | 100 | 100 | 100 | 100 | 95.6 | 78.3 | 48.7 | 32.7 | 21.6 | 14.8 | 11.4 | 9.1 | 7.0 |
| OGFC-20 | 100 | 100 | 95 | 57.1 | 16.5 | 10 | 6.1 | 5.5 | 5.2 | 4.9 | 4.5 | 4.1 | 3.3 |
| SUP-20 | 100 | 100 | 95.3 | 88.1 | 76.5 | 61.2 | 39.8 | 25.5 | 15.9 | 9.7 | 6.6 | 5.3 | 4.2 |
| SUP-25 | 100 | 96.7 | 83.1 | 75.9 | 67.3 | 55.2 | 35.9 | 23.4 | 14.3 | 8.5 | 5.8 | 4.8 | 3.7 |

**Table 7. SFM road performance test.**

| Basic performance | Test item | Evaluation index | Specimen size | Test temperature | Loading rate |
|---|---|---|---|---|---|
| High-temperature performance | Rutting test | Dynamic stability | 300mm×300mm×50mm | 60°C | - |
| Low-temperature performance | Three-point bending test | Flexural tensile strength, maximum flexural tensile strain | 250mm×30mm×35mm | -10°C | 50mm/min |
| Water stability | Freeze-thaw splitting test | Freeze-thaw splitting tensile strength ratio | 101.6mm×63.5mm (diameter × height) | 25°C | 50mm/min |

## 3 FE model construction

Asphalt pavement structure is a typical three-dimensional structure and is affected by multiple loads and material properties. It is difficult to obtain its mechanical response under temperature load coupling through traditional mechanical methods. Therefore, this paper uses the FE method to approximate the problem.

### 3.1 FE model of asphalt pavement

The temperature distribution along a road remains unchanged horizontally and vertically within a specific road area. Thus, to simplify the calculation, this article uses a two-dimensional model for analysis, with a model size of 3.75m × 3m, as shown in Fig 2. The road structure is set as a multi-layer structure, assuming that the inter-layer is continuous, and the thickness of each layer is shown in Table 8:

### 3.2 Load and boundary conditions

According to the Design Specification for Highway Asphalt Pavements (JTG D50-2017), the standard axle load refers to a single axle double wheelset with a load of 100kN, and the design parameters are shown in Table 9.

Huang Yangxian recommends the action form of road load as a rectangle plus two semicircles [19], as shown in Fig 3, which can be further simplified as a rectangular load. This article adopts the rectangular load form shown in Fig 4 [20], with a length of 192mm, a width of 184mm, a spacing of 135mm, and a load size of 0.7MPa. The corresponding load is shown in Fig 2 of the two-dimensional model.

The boundary condition is that the soil base is fully constrained, with lateral constraints on both sides. The specific boundary condition settings are shown in Fig 2.

### 3.3 Creep model

Asphalt mixture is a typical viscoelastic material and will produce creep deformation under load. Therefore, the creep characteristics of the asphalt surface should be considered when the FE modeling of road structure is carried out. In this paper, the time-hardening creep model was used to describe the creep behavior of asphalt mixture, as shown in Eq (4):

$$\dot{\varepsilon} = A\sigma^n t^m \tag{4}$$

where $\varepsilon$ is creep strain; $\sigma$ is stress; t is time; $A$, $n$, and $m$ are model parameters, which related to asphalt viscosity, aggregate maximum particle size, and aggregate angularity [21]. Fitting creep test data can obtain them, usually $A>0$, $n>0$, $-1<m\leq0$. The damping of materials used Rayleigh damping, in which the damping matrix is a linear combination of the mass matrix, as

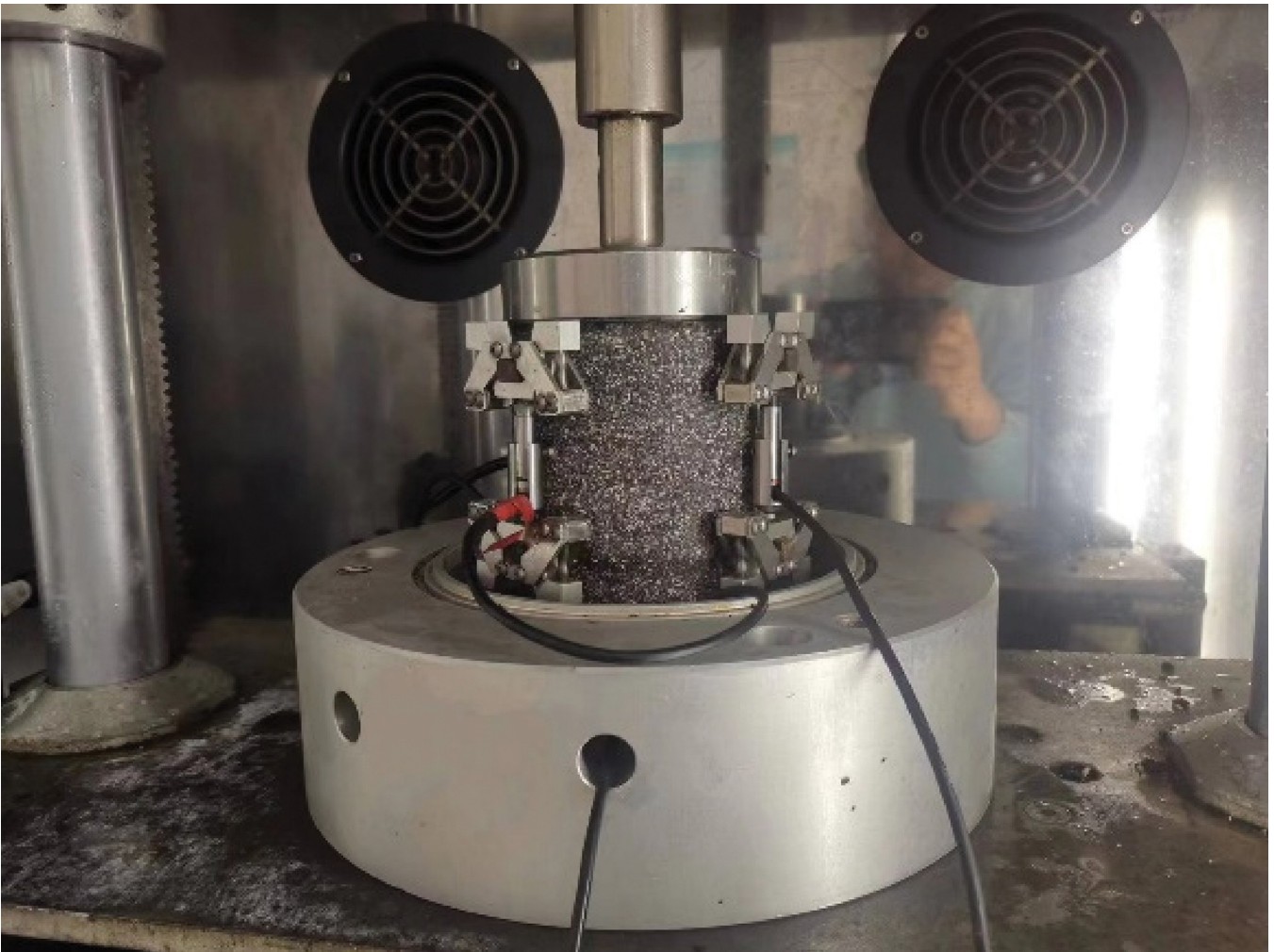

**Fig 1. Dynamic modulus test.**

shown in Eq (5):

$$\mathbf{C} = \alpha\mathbf{M} + \beta\mathbf{K} \tag{5}$$

where $\alpha$ and $\beta$ are constants defined according to material properties, $\beta$ of pavement material is small, and $\alpha$ can be directly entered in ABAQUS.

## 3.4 Establishment of pavement structure temperature field

The permanent deformation of asphalt pavement under vehicle load is greatly affected by temperature, so it is necessary to analyze the temperature distribution inside the pavement structure before modeling the rutting of asphalt pavement. The temperature field inside asphalt pavement is the sum of spatiotemporal temperature distribution. When the pavement structure is determined, the temperature field is mainly affected by many environmental factors such as solar radiation, effective sunshine time, daily maximum and minimum temperature, and wind speed.

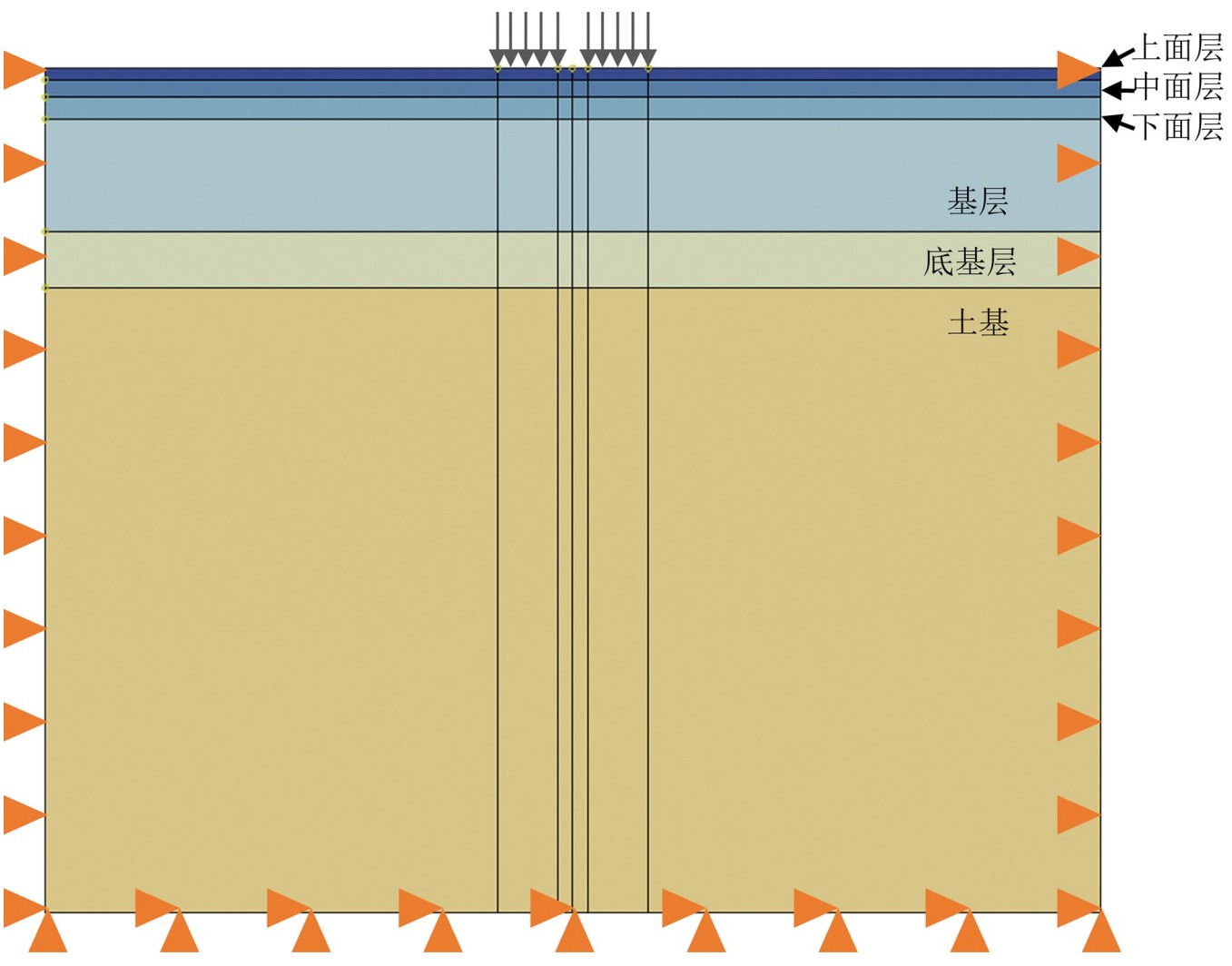

**Fig 2. Road FE model.**

**Table 8. Pavement structural layer materials.**

| Structural layer | Materials | Thickness /cm |
|---|---|---|
| Surface layer | asphalt concrete, Sup-13 | 4 |
| Middle layer | asphalt concrete, Sup-20 | 6 |
| Bottom layer | asphalt concrete, Sup-25 | 8 |
| Basic layer | Cement-stabilized macadam, CTB | 40 |
| Sub-base layer | Lime stabilized soil, LS | 20 |
| Soil foundation | Compacted soil, SG | - |

**Table 9. Design axle load parameters.**

| Design axle load | Tyre ground pressure (MPa) | Single wheel grounding equivalent circle diameter (mm) | Center distance between two wheels (mm) |
|---|---|---|---|
| 100 | 0.7 | 213.0 | 319.5 |

According to Baber's correlation theory, the periodic radiation passing through the sun can be approximated by Eq (6):

$$q(t) = \begin{cases} 0 & 0 \leq t < 12 - \dfrac{c}{12} \\ q_0 cosm\omega(t-12) & 12 - \dfrac{c}{12} \leq t \leq 12 + \dfrac{c}{12} \\ 0 & 12 + \dfrac{c}{12} < t \leq 24 \end{cases} \tag{6}$$

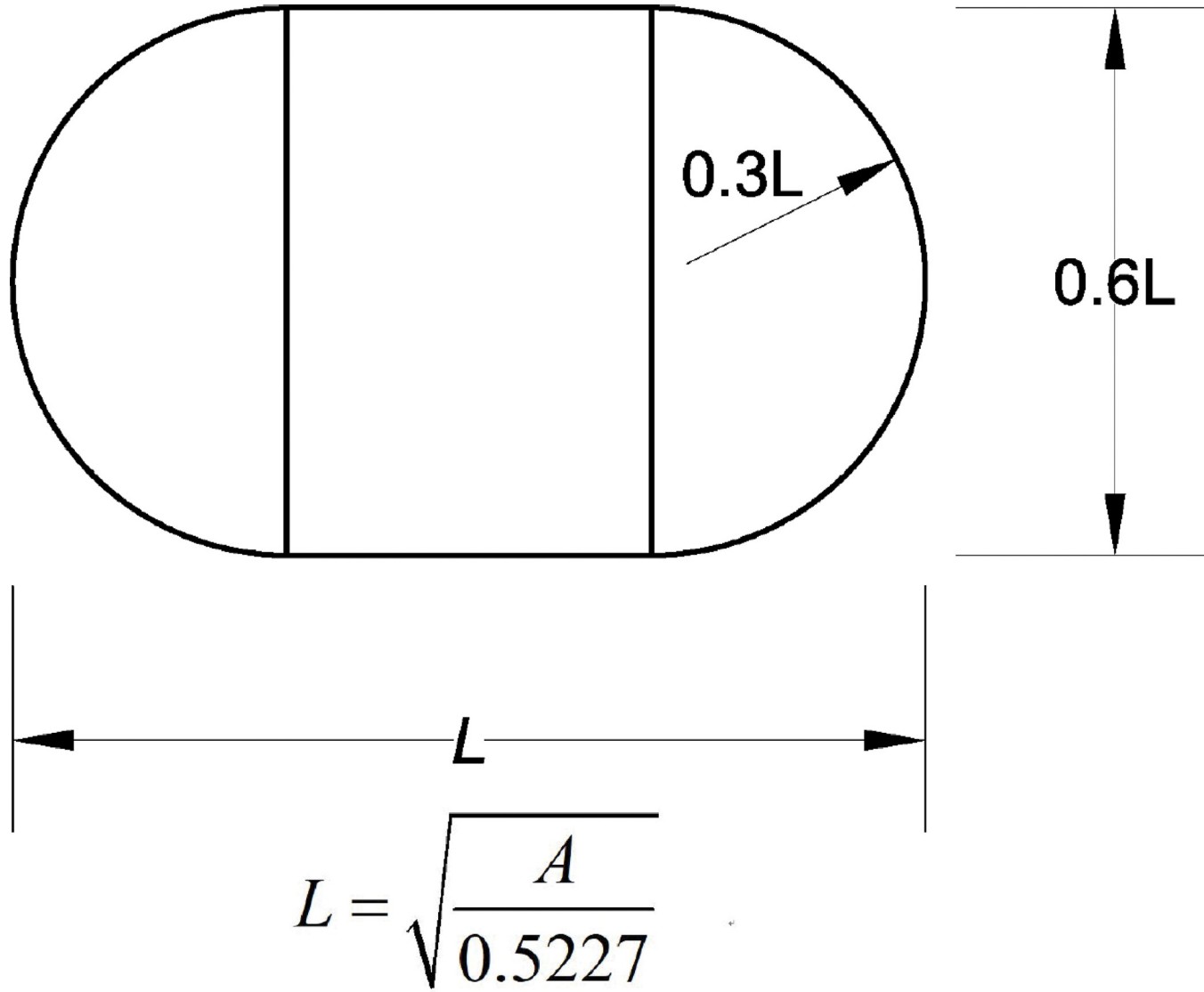

**Fig 3. Ground area of a tire.**

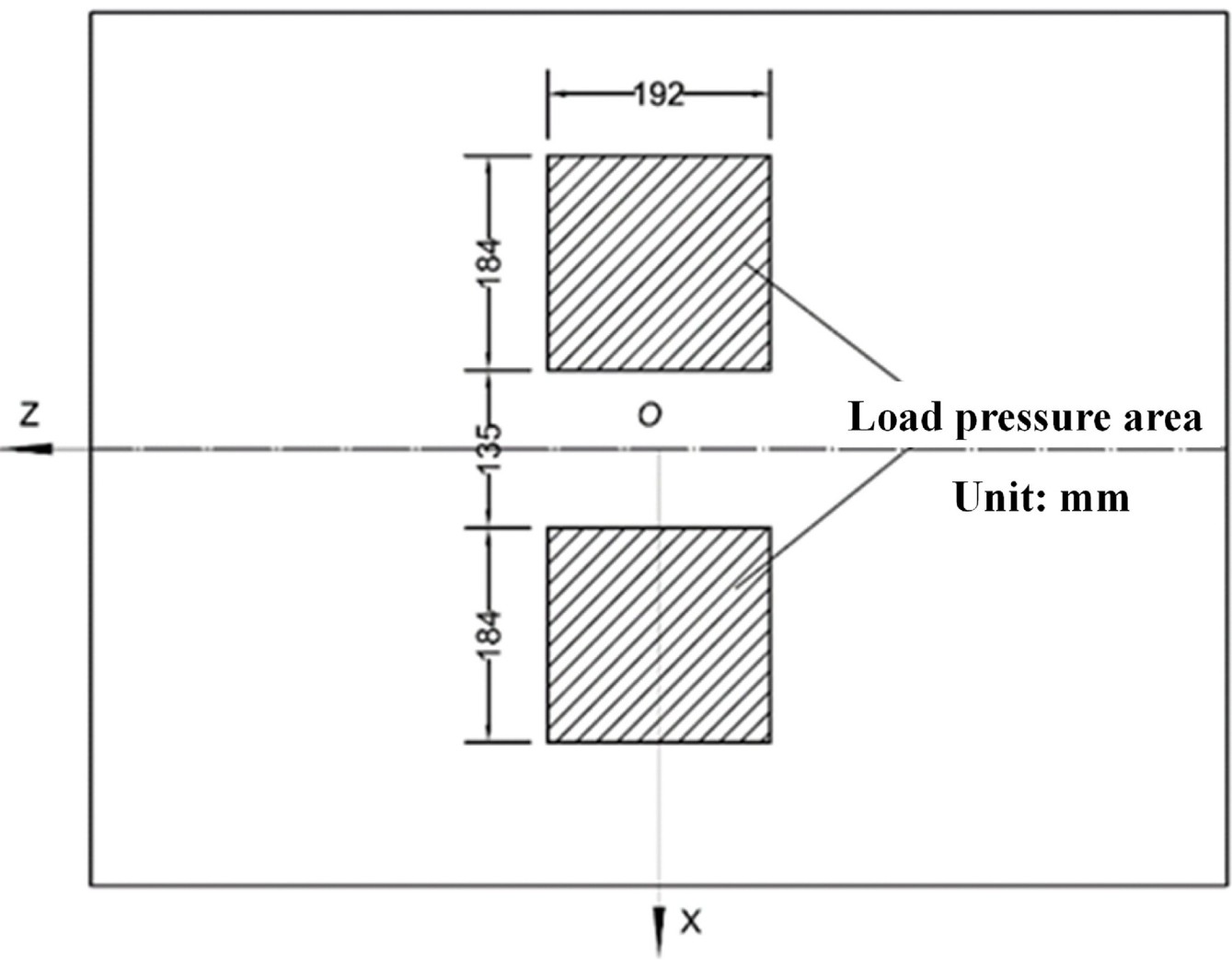

**Fig 4. Tire equivalent load area.**

where $q_0$ is the maximum radiation at noon, $q_0 = 0.131 \, \mathrm{m} \mathcal{Q}$, and $\mathcal{Q}$ is the total daily solar radiation, J/m2; $c$ is the actual effective sunshine hours, h; $\omega$ is the angular frequency, $\omega = 2\pi/24$, rad.

Eq (6) is a piecewise function that is difficult to operate when used. Therefore, researchers proposed a linear combination of two sine functions to approximate the periodic changes in atmospheric temperature:

$$T_a = \bar{T}_a + T_m[0.96sin\omega(t - t_0) + 0.14sin2\omega(t - t_0)] \tag{7}$$

where $\bar{T}_a$ is the daily average temperature, $\bar{T}_a = \frac{1}{2}\left(T_a^{max} + T_a^{min}\right)$, °C; $T_m$ is the daily temperature variation amplitude, $T_m = \frac{1}{2}\left(T_a^{max} - T_a^{min}\right)$, °C; $T_a^{max}$ and $T_a^{min}$ are the daily maximum and low temperature, respectively, °C; $\omega$ is the angular frequency, $\omega = 2\pi/24$, rad; $t_0$ is the initial phase.

There is a heat exchange phenomenon between the road surface and the atmosphere, which can be characterized by the heat exchange coefficient, as shown in Eq (8):

$$h_c = 3.7 v_w + 9.4 \tag{8}$$

where $h_c$ is the heat exchange coefficient, $W/(m^2 \cdot {}^\circ C)$; $v_w$ is the average daily wind speed, m/s.

The actual solar radiation received by the road surface is called effective radiation. Its size is related to the ground temperature, air temperature, cloud cover, air humidity, transparency, etc., given by Eq (9):

$$q_F = \epsilon\varsigma[(T_1|_{z=0} - T_Z)^4 - (T_a - T_Z)^4] \tag{9}$$

where $q_F$ is the ground effective radiation, $W/(m^2 \cdot {}^\circ C)$; $\epsilon$ is the emissivity of the road surface, and the asphalt road is 0.81; $\varsigma$ is the Stefan-Boltzmann constant, $\varsigma = 5.6697 \times 10^{-8}$, $W/(m^2 \cdot K^4)$; $T_1|_{z=0}$ is the road surface temperature, ${}^\circ C$; $T_a$ is the atmospheric temperature, ${}^\circ C$; $T_Z$ is the value of absolute zero, $T_Z = -273 {}^\circ C$.

Under the action of external heat sources, heat will be continuously transferred to the asphalt pavement. Such a heat transfer process can be characterized by the governing equations containing two basic laws: Fourier heat transfer law and energy conservation law. The governing equation used in this paper is the one-dimensional heat conduction formula, as shown in Eq (10):

$$\rho c \frac{\partial \theta}{\partial t} = k \frac{\partial^2 \theta}{\partial x^2} \tag{10}$$

where $\rho$ is the material density, kg/m3; $c$ is the specific heat of the material, $J/M/{}^\circ C$; $k$ is the thermal conductivity, $J/T/L/{}^\circ C$; $\theta$ is the heat, J; $x$ is the distance on the heat transfer path, m; $t$ is the time, h.

To obtain the temperature field of the pavement structure, we input the heat transfer parameters of each pavement layer in Eq (10) in the FE calculation, as shown in Table 10. Asphalt concrete and SFM thermal parameters were selected based on previous works [22, 23].

One day's temperature of a city in south China was selected as the external temperature condition, as shown in Fig 5.

## 3.5 Analysis of road rutting under the coupled action of temperature-load

Considering the creep characteristics of asphalt mixture, the internal strain of asphalt pavement under load can be calculated by Eq (11):

$$\varepsilon = \varepsilon_e + \varepsilon_p + \varepsilon_{ve} + \varepsilon_{vp} \tag{11}$$

where $\varepsilon$ is the total strain; $\varepsilon_e$ is recoverable elastic strain; $\varepsilon_p$ is unrecoverable plastic strain; $\varepsilon_{ve}$ is recoverable viscoelastic strain; $\varepsilon_{vp}$ is a non-recoverable viscoplastic strain.

**Table 10. Thermal attribute parameters of pavement temperature field analysis.**

| Parameter | Sup-13 | Sup-20 | Sup-25 | SFM | CTB | LS | SG |
|---|---|---|---|---|---|---|---|
| Thermal conductivity $k$ ($J/m \cdot h \cdot {}^\circ C$) | 4000 | 4000 | 4000 | 3700 | 5500 | 5000 | 5500 |
| Density $\rho$(kg/m³) | 2400 | 2400 | 2400 | 2500 | 2400 | 1900 | 1800 |
| Thermal capacity $c$(J/kg·°C) | 900 | 900 | 900 | 1000 | 920 | 940 | 1000 |
| Solar radiation absorption rate | 0.9 | | | | | | |
| Absolute zero $T_z$(°C) | -273 | | | | | | |
| Boltzmann constant $\varsigma$ ($J/(h \cdot m^2 \cdot k^4)$) | $2.041092 \times 10^{-4}$ | | | | | | |

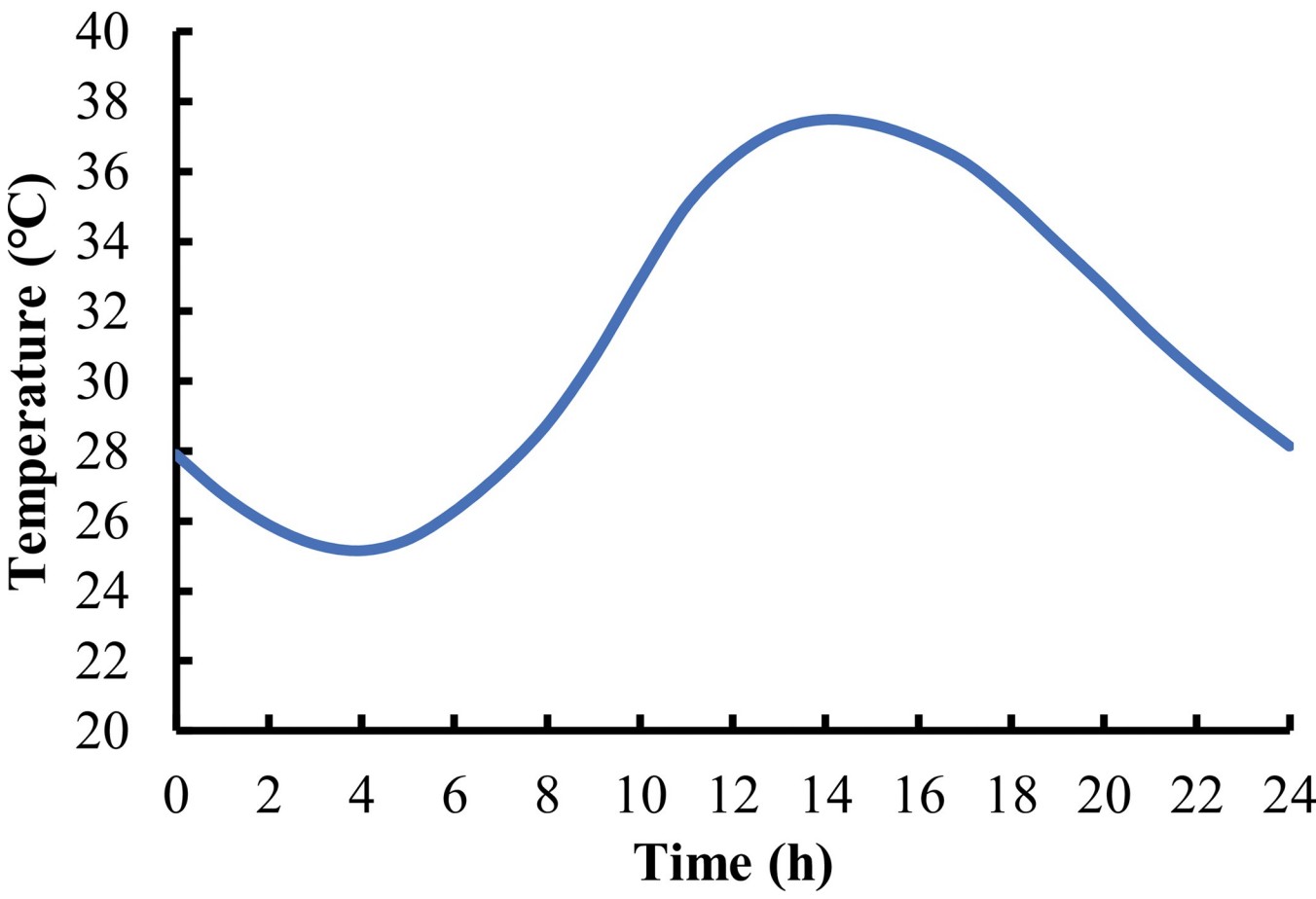

**Fig 5. Typical temperature changes in a day in cities in southern China.**

In general, the main contribution of road rutting comes from the viscoplastic deformation of asphalt mixture under long-term load. According to the relevant test methods in Section 1.4, the dynamic models of SUP-13, SUP-20, SUP-25, and SFM of pavement structure were tested at different temperatures. Previous works showed that the dynamic modulus changed from 1800MPa to 2000MPa with temperature changing from 60˚C to -10˚C [24, 25]. Based on the previous simulations, the most commonly used Poisson's ratio of SFM were 0.25 [26, 27] and 0.3 [28, 29]. Thus, we selected an ascending value from 0.25 to 0.3, changing with temperature as Poisson's ratio of SFM in this work. In addition, according to related research [21, 30, 31], the creep model (Eq (4)) was fitted by using the test data of the asphalt mixture. The final material parameters are shown in Table 11. Since there is no creeping phenomenon in the base and soil foundation, only the elastic parameters shown in Table 12 are taken.

The cumulative load action time can be calculated according to Eq (12):

$$t = \frac{0.36NP}{n_w pBv} \tag{12}$$

where $t$ is the cumulative action time of wheel load, s; $N$ is the number of wheel loads; $P$ is the axle weight of the vehicle, kN; $n_w$ is the number of wheels of the shaft; $p$ is the tire grounding pressure, MPa; $B$ is the tire ground width, cm; $v$ is the driving speed, km/h.

The daily flow of a river crossing passage in a specific city in southern China is about 249000 vehicles, with a driving speed of 80km/h. According to Eq (12), the cumulative load

**Table 11. Material parameters of asphalt mixture.**

| Mixture | Temperature (°C) | Elastic modulus (MPa) | Poisson's ratio | A | n | m |
|---|---|---|---|---|---|---|
| SUP-13 | 20 | 910.4 | 0.25 | $5.36 \times 10^{-11}$ | 0.925 | -0.611 |
| | 30 | 658.3 | 0.3 | $3.41 \times 10^{-9}$ | 0.834 | -0.582 |
| | 40 | 591.7 | 0.35 | $2.99 \times 10^{-8}$ | 0.769 | -0.547 |
| | 50 | 557.6 | 0.4 | $1.44 \times 10^{-6}$ | 0.395 | -0.515 |
| | 60 | 529.0 | 0.45 | $1.47 \times 10^{-5}$ | 0.328 | -0.496 |
| SUP-20 | 20 | 959.0 | 0.25 | $5.11 \times 10^{-11}$ | 0.942 | -0.598 |
| | 30 | 759.0 | 0.3 | $3.24 \times 10^{-9}$ | 0.844 | -0.566 |
| | 40 | 689.0 | 0.35 | $2.89 \times 10^{-8}$ | 0.779 | -0.541 |
| | 50 | 459.0 | 0.4 | $3.64 \times 10^{-6}$ | 0.535 | -0.506 |
| | 60 | 409.0 | 0.45 | $5.38 \times 10^{-5}$ | 0.399 | -0.472 |
| SUP-25 | 20 | 1009.0 | 0.25 | $4.89 \times 10^{-11}$ | 0.912 | -0.559 |
| | 30 | 829.0 | 0.3 | $3.56 \times 10^{-9}$ | 0.834 | -0.546 |
| | 40 | 734.0 | 0.35 | $2.44 \times 10^{-8}$ | 0.788 | -0.532 |
| | 50 | 501.0 | 0.4 | $1.67 \times 10^{-6}$ | 0.474 | -0.520 |
| | 60 | 409.0 | 0.45 | $4.79 \times 10^{-5}$ | 0.352 | -0.431 |
| SFM | 20 | 8836.6 | 0.25 | $4.66 \times 10^{-11}$ | 0.831 | -0.411 |
| | 30 | 5072.1 | 0.26 | $5.36 \times 10^{-10}$ | 0.774 | -0.395 |
| | 40 | 3344.9 | 0.27 | $7.85 \times 10^{-9}$ | 0.691 | -0.431 |
| | 50 | 2790.9 | 0.28 | $3.49 \times 10^{-8}$ | 0.664 | -0.455 |
| | 60 | 1864.0 | 0.3 | $5.11 \times 10^{-7}$ | 0.473 | -0.481 |

action time is about 1882 seconds. To more accurately reflect the load action situation at different periods of the day, the cumulative load action time per hour can be approximately obtained based on the distribution of traffic volume at different periods within 24 hours. The results are shown in Table 13.

According to the cumulative axle load action time in different periods of the day, as shown in Table 13, the rutting calculation of pavement structure under different temperatures and load coupling was carried out in this paper. In the FE simulation of road structure rutting under the coupling effect of temperature and load, the sequential coupling method was used, which involves conducting heat transfer analysis on the road structure first and then inputting the obtained temperature field in the form of field variables into the mechanical analysis to obtain the rutting changes at different times. In heat transfer analysis, the unit type is an 8-node quadrilateral heat transfer element (DC2D8). In mechanical analysis, the element type is an 8-node plane strain quadrilateral element (CPE8R).

## 3.6 Model verification

To verify the relevant models and calculation results in FE simulation, such as Eqs (12), (4), and the relevant parameters in Table 11, accelerated loading tests (In this study, a model mobile loading simulator (MMLS3, MLS Test Systems, Stellenbosch, South Africa)) were used to verify the load-deformation response of asphalt concrete at room temperature (25°C). The

**Table 12. Parameters of base and soil-base materials.**

| Structural layer | Elastic modulus (MPa) | Poisson's ratio |
|---|---|---|
| Basic layer CTB | 1300 | 0.20 |
| Sub-base layer LS | 900 | 0.25 |
| Soil foundation SG | 50 | 0.30 |

**Table 13. Cumulative load acting time of each period within 24h.**

| Time (h) | 1 | 2 | 3 | 4 | 5 | 6 | 7 | 8 |
|---|---|---|---|---|---|---|---|---|
| Traffic ratio (%) | 1.0193 | 0.8135 | 0.7311 | 0.4018 | 0.5791 | 0.8134 | 1.1427 | 2.0482 |
| Action time (s) | 19.18 | 15.31 | 13.76 | 7.56 | 10.90 | 15.31 | 21.51 | 38.54 |
| Time (h) | 9 | 10 | 11 | 12 | 13 | 14 | 15 | 16 |
| Traffic ratio (%) | 2.8714 | 4.5177 | 5.7525 | 7.3988 | 7.8104 | 6.5756 | 7.3988 | 8.6336 |
| Action time (s) | 54.04 | 85.02 | 108.26 | 139.24 | 146.98 | 123.75 | 139.24 | 162.48 |
| Time (h) | 17 | 18 | 19 | 20 | 21 | 22 | 23 | 24 |
| Traffic ratio (%) | 9.0452 | 7.8104 | 6.9872 | 4.9293 | 4.1060 | 3.6945 | 2.8714 | 2.0482 |
| Action time (s) | 170.22 | 146.98 | 131.49 | 92.76 | 77.27 | 69.53 | 54.04 | 38.54 |

test pieces measured 300×250×50mm and were loaded 1,000,000 times. The test pieces were taken out at regular intervals for rut depth measurement. The rubber tire had a contact pressure of 0.7MPa to the road surface, and the wheel ran at 7,200 revolutions/h. Due to the use of the same model for each layer of the asphalt mixture, the validation of Sup-13 and SFM was conducted separately (Fig 6).

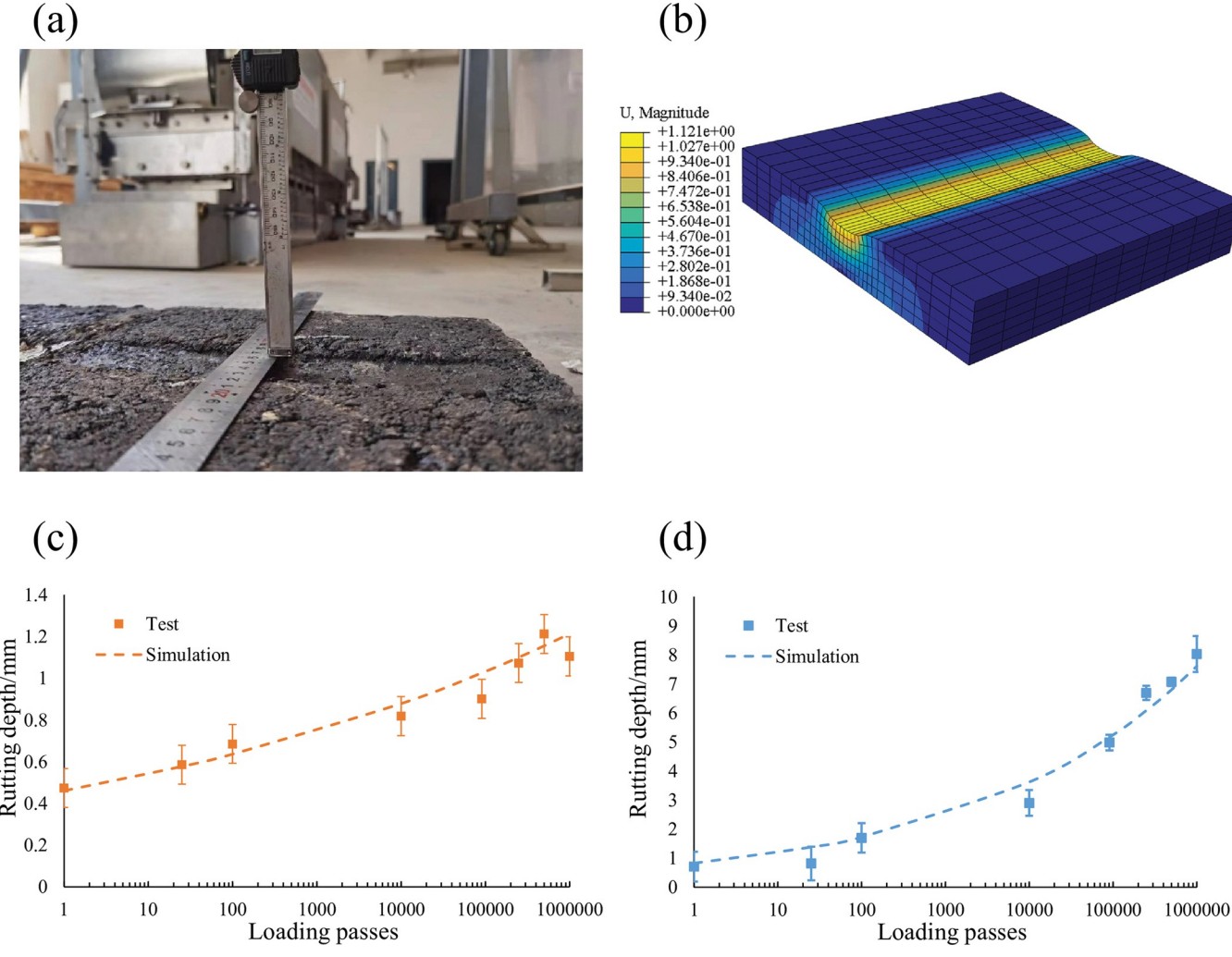

**Fig 6.** Accelerated loading verification test and simulation: (a) MMLS3 test; (b) FE simulation; (c) SFM validation; (d) SUP-13 validation.

A FE model was constructed and loaded under the same conditions. According to Eq (12), the corresponding number of axle loads was converted into the duration of the load. The comparison between the simulation results and the experimental results is shown in Fig 6(C) and 6 (D). The results show that the FE simulation is close to the experimental results and has high accuracy. The FE method proposed in this article can effectively predict the rutting depth of actual roads under temperature load coupling.

## 4 Results and discussion

### 4.1 Pavement structure temperature field

According to the relevant model and contents of the temperature field in Section 2.4, FE analysis was carried out on the temperature variation of road structure in one day. The obtained road structure temperature field was shown in Fig 7:

Fig 7 shows that the temperature field distribution in the pavement structure varies at different times of the day and with changes in environmental temperature. Compared to changes in temperature, the temperature change in the pavement structure has a certain lag. To more clearly demonstrate the temperature variation within the pavement structure over time, this paper derives the FE calculation results of temperature along the thickness direction of the pavement structure, as shown in Fig 8.

As shown in Fig 8, the temperature change shows a decreasing trend with the thickness, indicating that the farther away from the road surface, the less affected the temperature change. The pavement structure layer is subject to the greatest temperature change, so it is necessary to consider the influence of the temperature field of the pavement structure when conducting mechanical analysis. In addition, starting from 18 hours, the temperature in the middle of the road surface is higher than that of the road surface due to the weakening and disappearance of solar radiation after sunset, resulting in a decrease in road surface temperature. By comparing Figs 5 and 8, it can be found that at the highest temperature of 14 hours, the temperature inside the road structure is higher than the atmospheric temperature due to the high effective absorption rate of solar radiation on the road surface.

### 4.2 Rutting analysis of SFM at different levels

Based on the temperature field of the pavement structure and the creep characteristics of the asphalt mixture, this section conducted permanent deformation calculation of the pavement structure under temperature load coupling to explore the anti-rutting and adaptive layer of SFM. Fig 9 shows the cumulative deformation of the pavement structure under temperature and load coupling during different periods, including the calculation results of permanent deformation of the traditional pavement structure and the pavement surface with SFM located in the surface, middle, and bottom layers, respectively.

By comparing Fig 9(A) with 9(B), 9(C) and 9(D), it can be found that the vertical deformation of traditional asphalt pavement generates a maximum deformation of 5.4mm under the coupled action of temperature and load. It is the deepest rut among all four pavement structures, indicating that using SFM in pavement structures can reduce the rutting depth, which is also consistent with many relevant research results. In addition, by comparing Fig 9(B), 9(C) and 9(D), it can be found that the mechanical response changes with the distinct layers SFM located. The maximum deformation of 4.0mm, 1.3mm, and 4.4mm was observed when SFM was used in the top, middle, and bottom layers. Thus, locating SFM in the middle layer can produce the optimal anti-rutting effect. The formation of rutting is mainly due to the compressive stress and shear stress generated under the action of traffic load exceeding the strength of the road material, which can be roughly divided into compaction mode and dilative heave

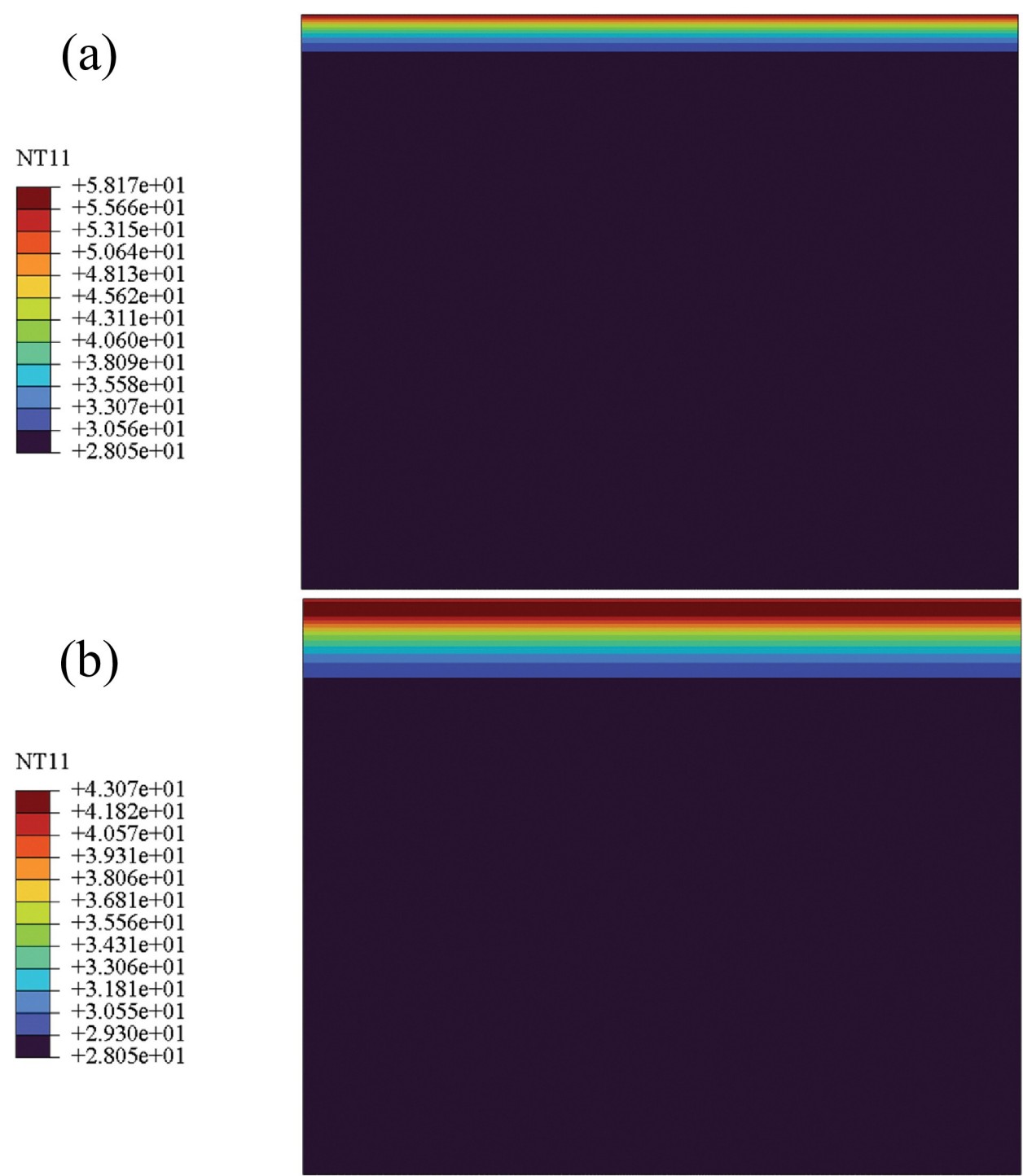

**Fig 7.** Temperature field of pavement structure in southern cities in one day: (a) noon; (b) 18 o'clock.

mode according to the different positions and direction of action. Therefore, to explore whether SFM will impact rutting types after being used in the pavement structure, this section outputs the displacement vector of pavement under the coupling action of temperature and load, as shown in Fig 10.

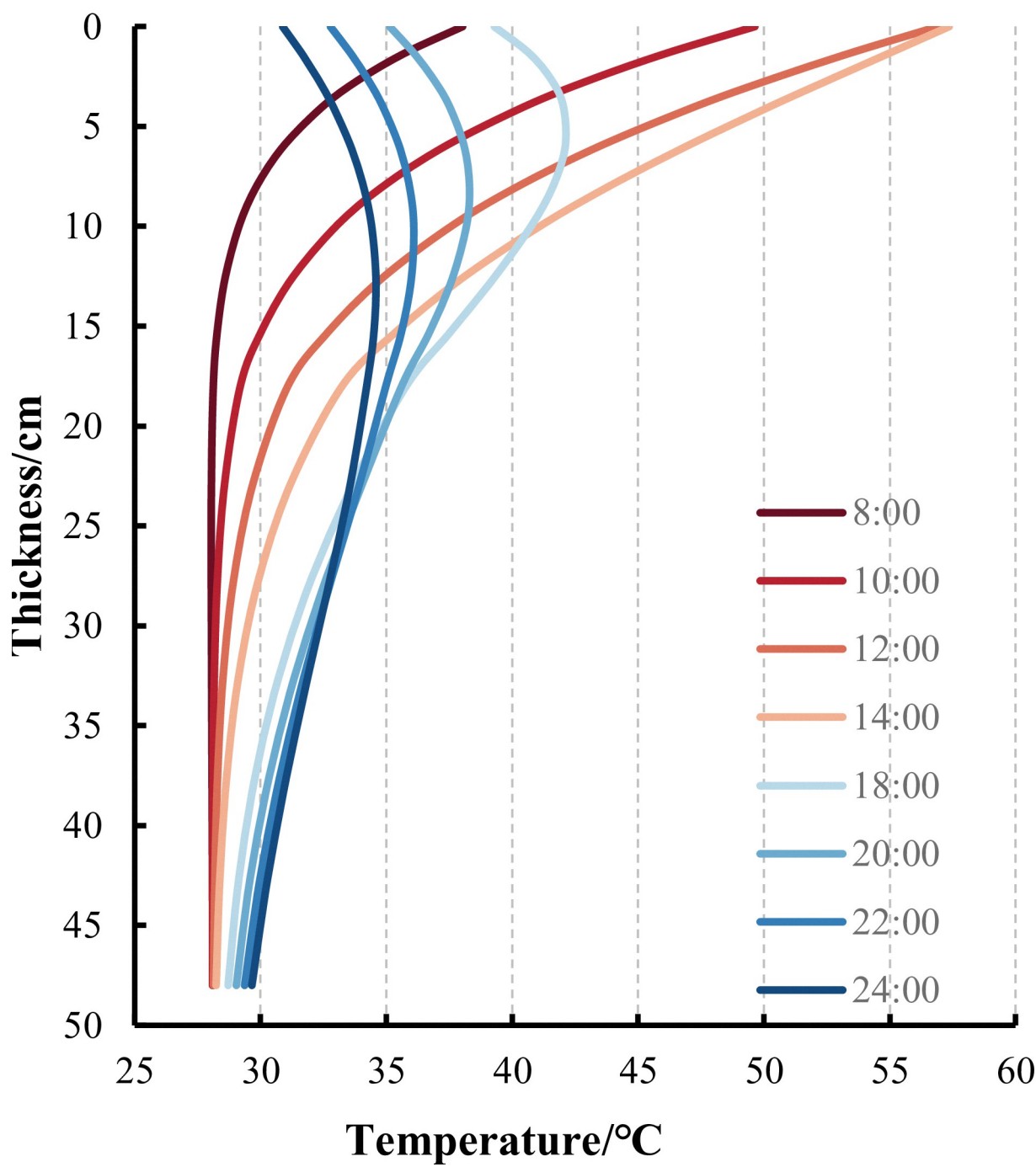

**Fig 8. Temperature distribution with thickness in road structure at different times of the day.**

By observing Fig 10, it can be observed that the traditional asphalt pavement (Fig 10(A)) and the pavement structure with SFM located in the surface and bottom layer (Fig 10(B) and 10(D)) exhibit a trend of dilation rutting under the coupling effect of temperature and load. It is due to the insufficient shear strength of the asphalt mixture itself under high temperatures,

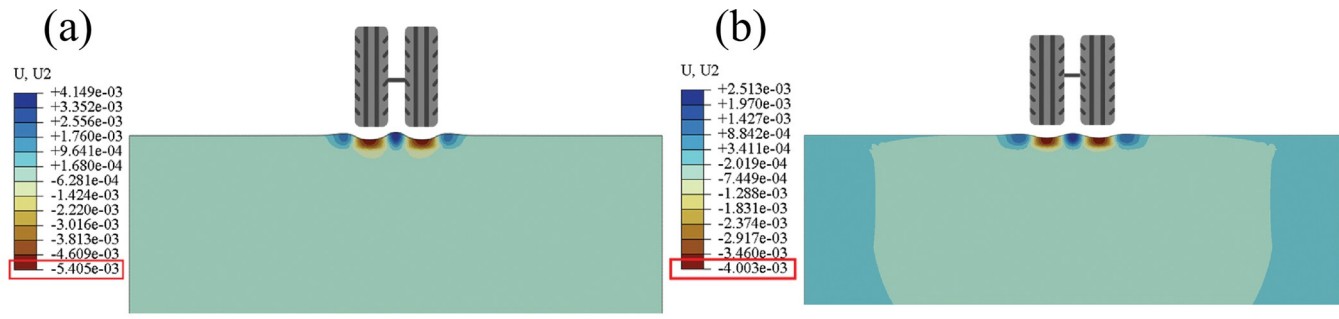

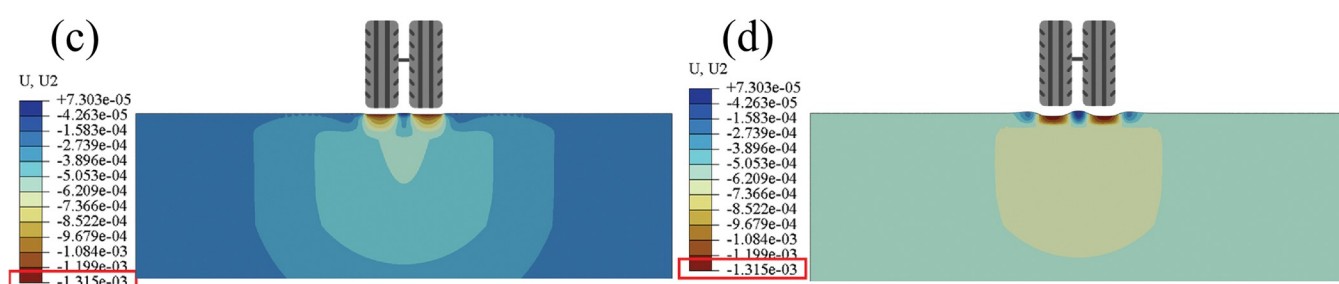

**Fig 9.** Vertical Permanent Deformation of Road Surface: (a) Asphalt Pavement; (b) SFM Surface Layer; (c) SFM Middle Layer; (d) SFM Bottom Layer.

resulting in lateral flow deformation of the surface layer. The pavement structure of SFM located in the middle layer (Fig 10(B)) tends to produce compacted rutting due to its high strength. When SFM is used in the middle layer, the pavement will not be heaved due to insufficient strength, only resulting in downward displacement. In addition, Fig 10(C) shows that SFM in the middle layer can distribute the load relatively more evenly into the beneath layers. However, when SFM is located in the other layers, the deformation is mainly concentrated in the middle layer, accelerating the fatigue failure of the asphalt mixture at that location and reducing the overall life of the road. Therefore, combining Figs 9 and 10, using SFM as the middle layer, will have the optimal effect on the overall anti-rutting effect of the road surface. For exploring the evolution of road surface ruts at different periods of the day, four pavement structures were output with different rut patterns at different periods, as shown in Fig 11.

Fig 11 shows four different pavement structures' cumulative rutting deformation patterns during several characteristic periods of 24 hours daily. It can be observed that the rutting depth deepens with time. Among them, the traditional asphalt pavement structure (Fig 11(A)) and the SFM pavement structure located in the surface and bottom layer (Fig 11(B) and 11 (D)) have relatively similar rutting morphology evolution. All show a dilative deformation trend starting from noon (rut depth greater than 0). Using SFM in the middle layers (Fig 11 (C)), there is a significant difference in the evolution of the rut morphology of the pavement structure compared to other pavement structures, which is a compaction trend (rut depth is less than 0). The compaction mode is self-stabilizing, causing the pavement to stiffen and thereby spread wheel load better, and a better load spreading leads to reduced stress on the subgrade, thereby reducing the risk and amount of rutting at that level.

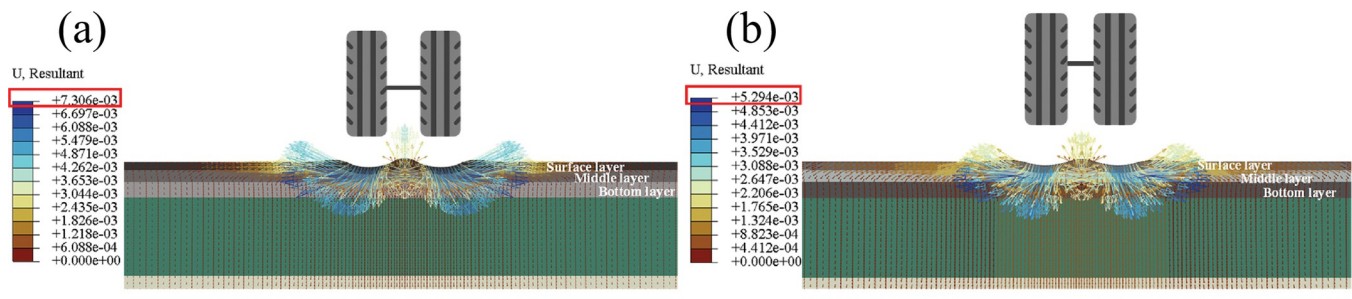

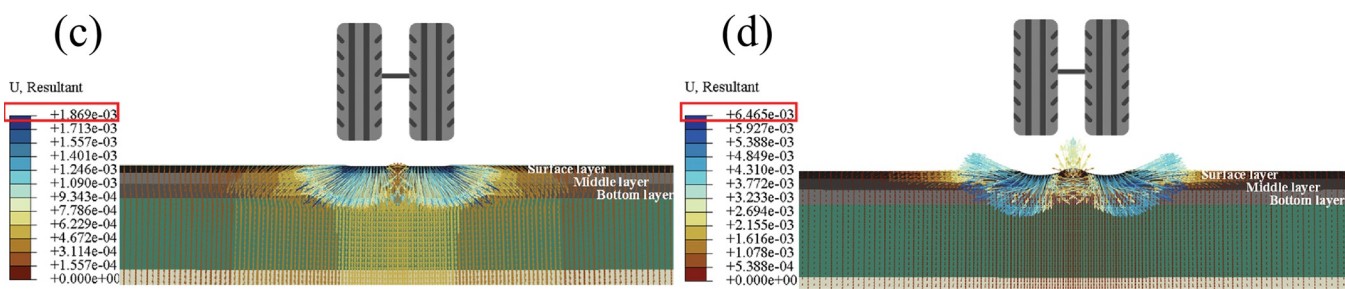

**Fig 10.** Road surface displacement vector: (a) Asphalt pavement; (b) SFM Surface Layer; (c) SFM Middle Layer; (d) SFM Bottom Layer.

The rutting of all pavement structures with or without SFM mainly occurs between 12:00 and 18:00. Based on the temperature inside the pavement shown in Fig 8, the appearance of rutting is associated with the temperature in the middle layer of the pavement structure. Although the modulus of SFM also varies with temperature, its overall strength is relatively high, so it can resist the coupling effect of temperature load when it is located in the middle layer.

## 4.3 The applicability of SFM under overload

Overloading is the main cause of road damage in China. According to previous studies, the maximum overload of vehicles can reach 200% of the standard load, and the tire pressure of heavy trucks exceeds 1.0MPa [24]. Thus, this section simulates rut formation under overload. The standard pressure that a tire applied on the pavement is P = 0.7MPa, and the overload conditions of 1.5P and 2P are selected in this section to calculate the anti-rut applicability of SFM under overload.

Absolute rutting refers to the maximum value (vertical displacement value) of the downward depression on the road surface. Fig 12(A)–12(C) show the absolute rutting of four different pavement structures under overload conditions. The results show that applying SFM to the middle layer can reduce absolute rutting on the road surface. Overloading does not change the cross-sectional geometry of the road surface rutting but only increases the absolute rutting depth. Relative rutting refers to the difference between the maximum depression and maximum uplift on the road surface. Fig 12(D) shows the relative rutting changes of four different pavement structures under overload conditions. The results show that locating SFM in the middle layer produces a small relative rutting depth under standard and overload conditions.

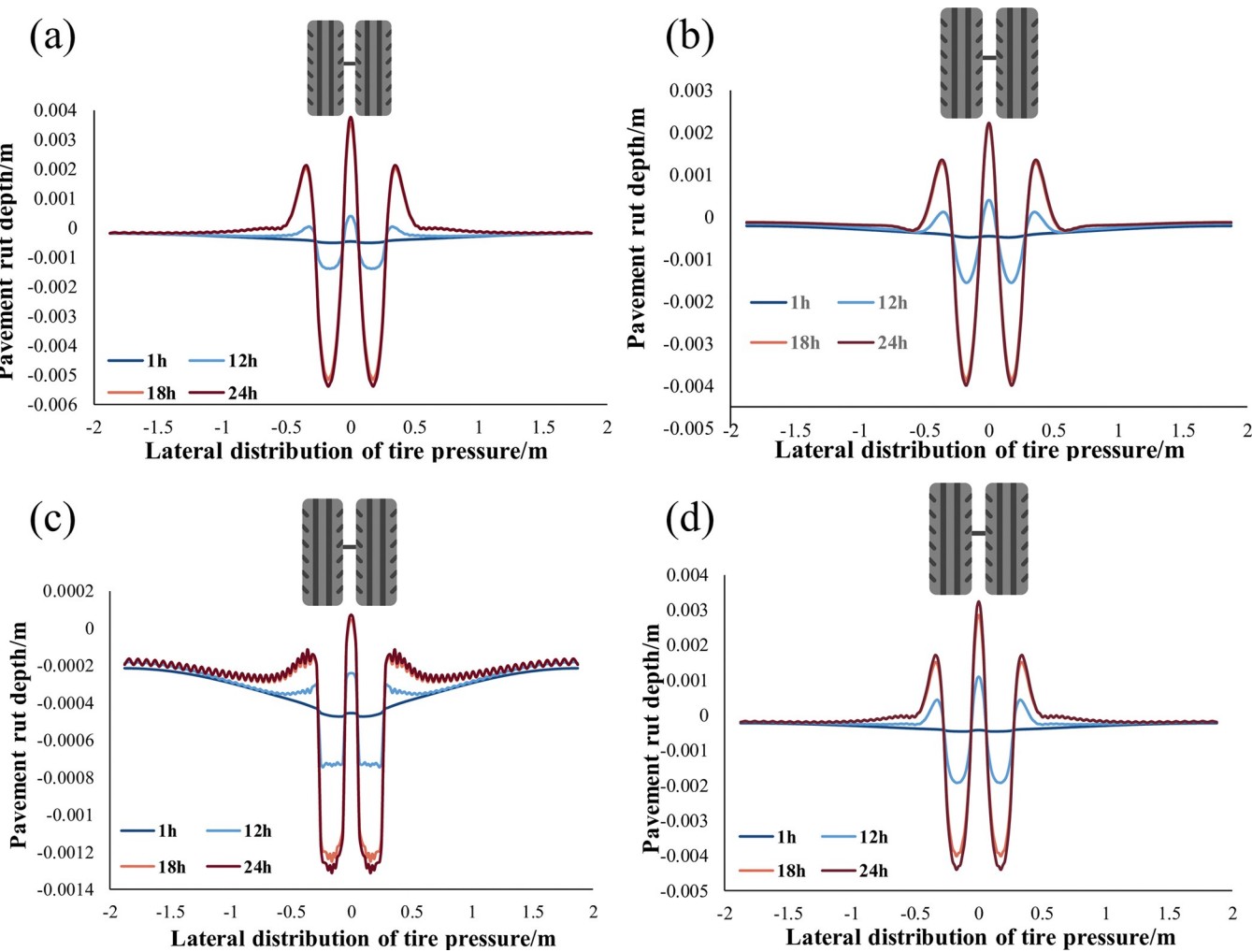

**Fig 11.** Changes in Road Surface Ruts in Southern Regions with Different Pavement Structures in One Day: (a) Asphalt Pavement; (b) SFM Surface Layer; (c) SFM Middle Layer; (d) SFM Bottom Layer.

Additionally, locating SFM in the middle and bottom layers reduces the growth of relative rut depth with the increasing load. Locating SFM in the top layer can reduce relative rutting depth due to its high strength. However, the weak strength of the beneath layers (asphalt mixture) and insufficient structural integrity cause the load to increase significantly with the overload (58% under 2P action). Therefore, considering the absolute and relative rutting values and the resistance to overloading, the application of SFM in the middle surface layer can better exert its anti-rutting function.

## 5 Conclusion

SFM is a novel construction material exhibiting both characteristics of cement and asphalt, increasing structural stiffness and reducing rutting. Previous studies revealed that the temperature load coupling effect is one of the leading causes of road rutting. However, few researchers focused on the anti-rutting impact and structural layer applicability of SFM under this effect. To address this issue, we developed a temperature-mechanical approach, building a FE model of the pavement structure method to simulate the rutting of SFM at different layers and times

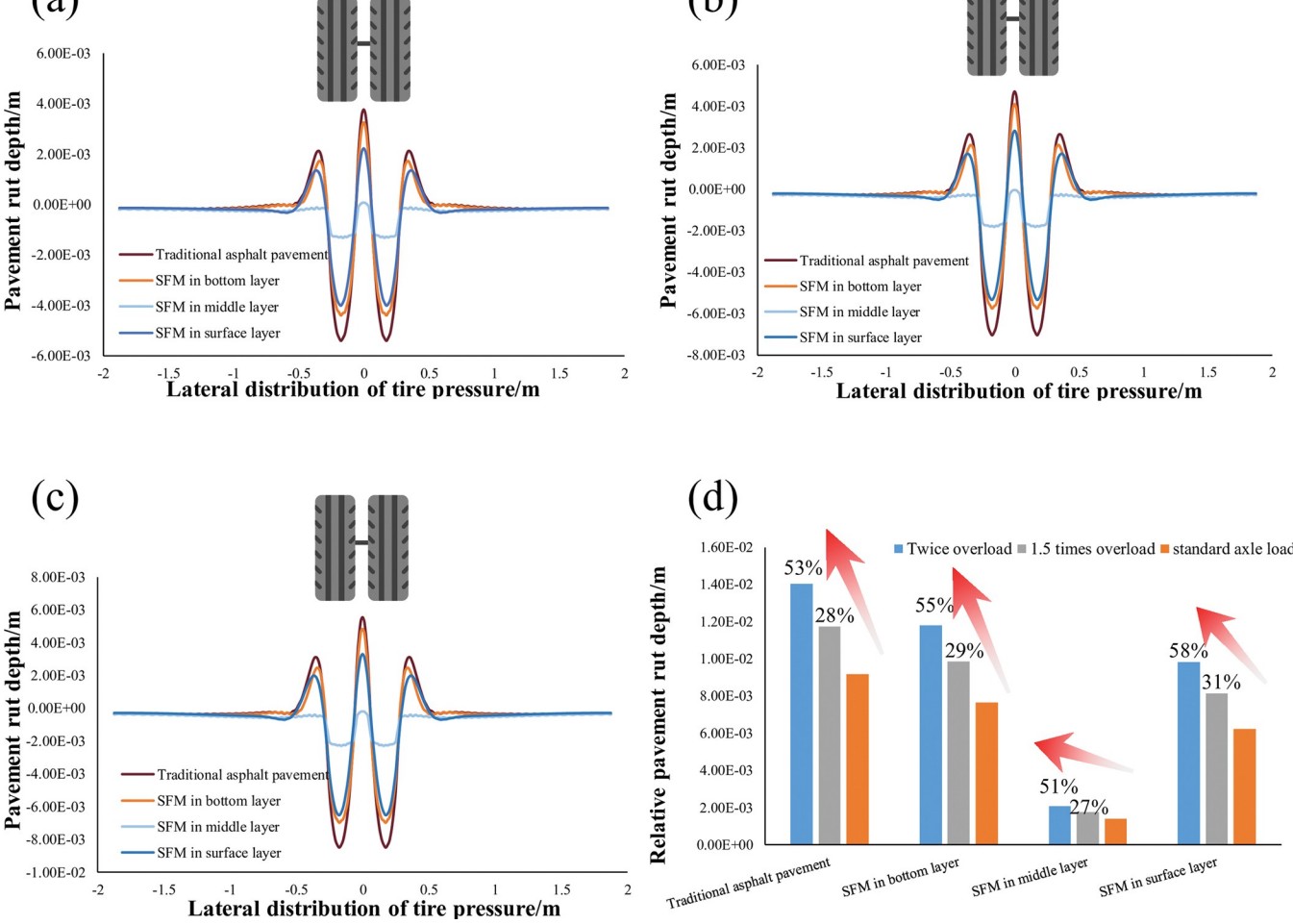

**Fig 12.** Four types of road structure rutting under overload: (a) rutting depth under 1P action; (b) rutting depth under 1.5P action; (c) Rut depth under the action of 2.0P; (d) Relative rutting depth varies with load.

of the day. An accelerated pavement test (APT) based validation indicated that the simulation results were accurate. Based on the presented investigation, the following conclusions were reached:

1. Compared with the fluctuation of air temperature, the temperature distribution in the pavement structure has a certain lag in a day. From 8:00 to 14:00, the temperature of the road surface structure gradually decreases from top to bottom. Starting from 18:00, the temperature in the middle of the road surface is higher than that of the road surface due to the weakening and disappearance of solar radiation after sunset, resulting in a decrease in road surface temperature.

2. The traditional asphalt pavement structure and the pavement structure with SFM in the surface and bottom layers have the upward and lateral displacement trend on the road surface under the coupling action of temperature and load, so they are typically unstable ruts. The road surface with SFM in the middle layer has no upward and lateral displacement trend, manifested as compacted ruts.

3. When SFM is in the surface, middle, and bottom layers, the pavement generates maximum deformation of 4.0mm, 1.3mm, and 4.4mm under the coupling effect of temperature and load in a day. The daily rutting of pavement with or without SFM mainly occurs between 12:00 and 18:00. Using SFM in the middle layer can distribute the load relatively more evenly into the beneath layers, and the pavement will not be heaved due to insufficient strength.

4. Locating SFM in the middle layer of pavement gives a better resistance to overloading. Compared with the standard load, the relative rutting increase under two times overload is only 51%. Considering the rutting depth, form, and resistance to rutting, the pavement structure with SFM in the middle surface layer has a better anti-rutting function.

## Supporting information

**S1 File. Data for Fig 5.**
(XLSX)

**S2 File. Data for Fig 6.**
(XLSX)

**S3 File. Data for Fig 8.**
(XLSX)

**S4 File. Data for Fig 11.**
(XLSX)

**S5 File. Data for Fig 12.**
(XLSX)

## Author Contributions

**Conceptualization:** Maohua Yu, Minda Ren.

**Funding acquisition:** Maohua Yu, Tianming He, Kejian Xu.

**Investigation:** Maohua Yu, Tianming He, Kejian Xu.

**Methodology:** Minda Ren.

**Resources:** Kejian Xu, Hong Cheng.

**Software:** Hong Cheng.

**Validation:** Minda Ren.

**Writing – original draft:** Tianming He, Kejian Xu, Minda Ren.

**Writing – review & editing:** Tianming He, Minda Ren.

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
