## [Decision Letter · Decision Letter 0]

23 Jul 2023

PONE-D-23-21160Structural Layer Applicability of Semi-Flexible Material for Rutting Resistance: A Coupled Temperature-mechanical ApproachPLOS ONE

Dear Dr. Ren,

Thank you for submitting your manuscript to PLOS ONE. After careful consideration, we feel that it has merit but does not fully meet PLOS ONE’s publication criteria as it currently stands. Therefore, we invite you to submit a revised version of the manuscript that addresses the points raised during the review process.

We look forward to receiving your revised manuscript.

Kind regards,

Khalil Abdelrazek Khalil, Ph.D.

Academic Editor

PLOS ONE

“The authors acknowledge the support received from the Jiangxi Provincial Department of Transportation Project (NO. 2022SF003).”

“The authors acknowledge the support received from the Jiangxi Provincial Department of Transportation Project (NO. 2022SF003).”

“The authors acknowledge the support received from the Jiangxi Provincial Department of Transportation Project (NO. 2022SF003).”

5. We note that Figures 1 and 6 in your submission contain copyrighted images. All PLOS content is published under the Creative Commons Attribution License (CC BY 4.0), which means that the manuscript, images, and Supporting Information files will be freely available online, and any third party is permitted to access, download, copy, distribute, and use these materials in any way, even commercially, with proper attribution. For more information, see our copyright guidelines: http://journals.plos.org/plosone/s/licenses-and-copyright.

1. You may seek permission from the original copyright holder of Figures 1 and 6 to publish the content specifically under the CC BY 4.0 license.

Reviewers' comments:

Reviewer's Responses to Questions

**Comments to the Author**

1. Is the manuscript technically sound, and do the data support the conclusions?

Reviewer #1: Yes

Reviewer #2: Yes

2. Has the statistical analysis been performed appropriately and rigorously? 

Reviewer #1: Yes

Reviewer #2: Yes

3. Have the authors made all data underlying the findings in their manuscript fully available?

Reviewer #1: Yes

Reviewer #2: Yes

4. Is the manuscript presented in an intelligible fashion and written in standard English?

Reviewer #1: Yes

Reviewer #2: Yes

5. Review Comments to the Author

Reviewer #1: The Semi-flexible material (SFM) may be have a performance of rutting resistance. But in summer, asphalt mixture may be in soft state, and cement is a fragile material. So this structure may be cracked because of there material performance.

Reviewer #2: The Structural Layer Applicability of Semi-Flexible Material for Rutting Resistance was researched. The coupled temperature-mechanical approach model was instituted by Abaqus. The results are significant. However, the simulation process still needs to be optimized.

1. The semi-flexible pavement material combines the rigidity of the cement-based grout. So it exhibits incomplete elasticity. The properties of completely elastic materials are different from those of actual materials. The simulated parameters may not be reasonable.

2. In Table 11, the creep parameters of SFM are missing. The creep parameters of SFM are suggested to be investigated by experiments. SFM should also have creep parameters.

3. Same mistakes: a) The line number is missing b) The acronym SFP in Table 11 is not defined.

6. PLOS authors have the option to publish the peer review history of their article (what does this mean?). If published, this will include your full peer review and any attached files.

Reviewer #1: **Yes: **Xiaoming HUANG

Reviewer #2: No

---

## [Author Response · Author response to Decision Letter 0]

15 Aug 2023

Reviewer #1: The Semi-flexible material (SFM) may be have a performance of rutting resistance. But in summer, asphalt mixture may be in soft state, and cement is a fragile material. So this structure may be cracked because of there material performance.

Response: There are still some distresses in pavement besides rutting, among which cracking is significant. Based on previous studies, the compatibility among components (aggregate, asphalt, and grouting) in SFM is a primary issue. It is an interesting topic, which we will investigate in future works.

Reviewer #2: The Structural Layer Applicability of Semi-Flexible Material for Rutting Resistance was researched. The coupled temperature-mechanical approach model was instituted by Abaqus. The results are significant. However, the simulation process still needs to be optimized.

1. The semi-flexible pavement material combines the rigidity of the cement-based grout. So it exhibits incomplete elasticity. The properties of completely elastic materials are different from those of actual materials. The simulated parameters may not be reasonable.

Response: We assumed the SFM is a semi-flexible material and used a creep model to describe such incomplete elasticity. Further, to validate the creep model, we conducted the accelerated loading test to compare the measured rutting depth and the calculation one in the FE simulation. The results showed little difference between calculated and measured rutting depth. It implies that using the creep model with calibrated parameters can accurately predict the rutting depth of actual roads under temperature load coupling. 

2. In Table 11, the creep parameters of SFM are missing. The creep parameters of SFM are suggested to be investigated by experiments. SFM should also have creep parameters.

Response: We added the creep parameters of SFM in Table 11.

3. Same mistakes: a) The line number is missing b) The acronym SFP in Table 11 is not defined.

Response: We added the line number. We also checked the whole article and changed the “SFP” to “SFM”.

---

## [Decision Letter · Decision Letter 1]

7 Sep 2023

PONE-D-23-21160R1Structural Layer Applicability of Semi-Flexible Material for Rutting Resistance: A Coupled Temperature-mechanical ApproachPLOS ONE

Dear Dr. Ren,

Thank you for submitting your manuscript to PLOS ONE. After careful consideration, we feel that it has merit but does not fully meet PLOS ONE’s publication criteria as it currently stands. Therefore, we invite you to submit a revised version of the manuscript that addresses the points raised during the review process. Please submit your revised manuscript by Oct 22 2023 11:59PM. If you will need more time than this to complete your revisions, please reply to this message or contact the journal office at plosone@plos.org. Please include the following items when submitting your revised manuscript:A rebuttal letter that responds to each point raised by the academic editor and reviewer(s). You should upload this letter as a separate file labeled 'Response to Reviewers'.A marked-up copy of your manuscript that highlights changes made to the original version. You should upload this as a separate file labeled 'Revised Manuscript with Track Changes'.An unmarked version of your revised paper without tracked changes. You should upload this as a separate file labeled 'Manuscript'.If applicable, we recommend that you deposit your laboratory protocols in protocols.io to enhance the reproducibility of your results. Protocols.io assigns your protocol its own identifier (DOI) so that it can be cited independently in the future. For instructions see: https://journals.plos.org/plosone/s/submission-guidelines#loc-laboratory-protocols. Additionally, PLOS ONE offers an option for publishing peer-reviewed Lab Protocol articles, which describe protocols hosted on protocols.io. Read more information on sharing protocols at https://plos.org/protocols?utm_medium=editorial-email&utm_source=authorletters&utm_campaign=protocols.

We look forward to receiving your revised manuscript.

Kind regards,

Khalil Abdelrazek Khalil, Ph.D.

Academic Editor

PLOS ONE

Journal Requirements:

Reviewers' comments:

Reviewer's Responses to Questions

**Comments to the Author**

1. If the authors have adequately addressed your comments raised in a previous round of review and you feel that this manuscript is now acceptable for publication, you may indicate that here to bypass the “Comments to the Author” section, enter your conflict of interest statement in the “Confidential to Editor” section, and submit your "Accept" recommendation.

Reviewer #2: All comments have been addressed

2. Is the manuscript technically sound, and do the data support the conclusions?

Reviewer #2: Yes

3. Has the statistical analysis been performed appropriately and rigorously? 

Reviewer #2: Yes

4. Have the authors made all data underlying the findings in their manuscript fully available?

Reviewer #2: Yes

5. Is the manuscript presented in an intelligible fashion and written in standard English?

Reviewer #2: Yes

6. Review Comments to the Author

Reviewer #2: The paper has undergone necessary rectifications. However, the following aspects require further discussion:

1. In relation to parameter selection, the thermal parameters and Poisson's ratio for the SFM material were aligned with those of conventional composite materials. It is crucial to elaborate on the source of these parameters—whether they were determined through empirical measurements or drawn from references. It is important to note the substantial influence of these parameters on the simulation outcomes.

2. Furthermore, a need for reconfirmation arises concerning the values of dynamic elastic moduli presented in Table 11. The provided data values do not align with the dynamic modulus values; instead, they bear a closer resemblance to static rebound moduli. Additionally, the extent of variation in the dynamic modulus of the SFM material in response to temperature would not exhibit such negligible values. Thus, a thorough reevaluation of the data's accuracy is of utmost importance.

7. PLOS authors have the option to publish the peer review history of their article (what does this mean?). If published, this will include your full peer review and any attached files.

Reviewer #2: No

---

## [Author Response · Author response to Decision Letter 1]

22 Sep 2023

Reviewer #2: The paper has undergone necessary rectifications. However, the following aspects require further discussion:

1. In relation to parameter selection, the thermal parameters and Poisson's ratio for the SFM material were aligned with those of conventional composite materials. It is crucial to elaborate on the source of these parameters—whether they were determined through empirical measurements or drawn from references. It is important to note the substantial influence of these parameters on the simulation outcomes.

Response: We detailed the parameter selection in lines 205-206 and 219-223, including the thermal parameters and Poisson's ratio for the SFM. These parameters were drawn from previous works, and the relative references were also cited. 

2. Furthermore, a need for reconfirmation arises concerning the values of dynamic elastic moduli presented in Table 11. The provided data values do not align with the dynamic modulus values; instead, they bear a closer resemblance to static rebound moduli. Additionally, the extent of variation in the dynamic modulus of the SFM material in response to temperature would not exhibit such negligible values. Thus, a thorough reevaluation of the data's accuracy is of utmost importance.

Response: We checked the data in Table 11 and modified the dynamic modulus values. Also, the results were recalculated using the improved parameters. Relevant content based on the simulation results were updated in Abstract, Results and Discussion, and Conclusion (line 306-309, line 322-329, line 339-345, line 349-351, line 366-367, line 371-372, line 374-377, and line 400-408).

---

## [Decision Letter · Decision Letter 2]

7 Nov 2023

Structural Layer Applicability of Semi-Flexible Material for Rutting Resistance: A Coupled Temperature-mechanical Approach

PONE-D-23-21160R2

Dear Dr. Ren,

We’re pleased to inform you that your manuscript has been judged scientifically suitable for publication and will be formally accepted for publication once it meets all outstanding technical requirements.

Kind regards,

Khalil Abdelrazek Khalil, Ph.D.

Academic Editor

PLOS ONE

Additional Editor Comments (optional):

Reviewers' comments:

Reviewer's Responses to Questions

**Comments to the Author**

1. If the authors have adequately addressed your comments raised in a previous round of review and you feel that this manuscript is now acceptable for publication, you may indicate that here to bypass the “Comments to the Author” section, enter your conflict of interest statement in the “Confidential to Editor” section, and submit your "Accept" recommendation.

Reviewer #2: All comments have been addressed

2. Is the manuscript technically sound, and do the data support the conclusions?

Reviewer #2: Yes

3. Has the statistical analysis been performed appropriately and rigorously? 

Reviewer #2: Yes

4. Have the authors made all data underlying the findings in their manuscript fully available?

Reviewer #2: Yes

5. Is the manuscript presented in an intelligible fashion and written in standard English?

Reviewer #2: Yes

6. Review Comments to the Author

Reviewer #2: The authors revised the paper carefully, now there aren't any other further revised comments, suggest to accept it.

7. PLOS authors have the option to publish the peer review history of their article (what does this mean?). If published, this will include your full peer review and any attached files.

Reviewer #2: No

---

## [Editor Report · Acceptance letter]

22 Nov 2023

PONE-D-23-21160R2 

Structural Layer Applicability of Semi-Flexible Material for Rutting Resistance: A Coupled Temperature-mechanical Approach 

Dear Dr. Ren:

I'm pleased to inform you that your manuscript has been deemed suitable for publication in PLOS ONE. Congratulations! Your manuscript is now with our production department. 

Kind regards, 

on behalf of

Dr. Khalil Abdelrazek Khalil 

Academic Editor

PLOS ONE